# *Arabidopsis* 14-3-3 epsilon members contribute to polarity of PIN auxin carrier and auxin transport-related development

**Jutta Keicher, Nina Jaspert, Katrin Weckermann, Claudia Möller, Christian Throm, Aaron Kintzi, Claudia Oecking\***

Plant Physiology, Center for Plant Molecular Biology, University of Tübingen, Tübingen, Germany

**Abstract** Eukaryotic 14-3-3 proteins have been implicated in the regulation of diverse biological processes by phosphorylation-dependent protein-protein interactions. The *Arabidopsis* genome encodes two groups of 14-3-3s, one of which – epsilon – is thought to fulfill conserved cellular functions. Here, we assessed the in vivo role of the ancestral 14-3-3 epsilon group members. Their simultaneous and conditional repression by RNA interference and artificial microRNA in seedlings led to altered distribution patterns of the phytohormone auxin and associated auxin transport-related phenotypes, such as agravitropic growth. Moreover, 14-3-3 epsilon members were required for pronounced polar distribution of PIN-FORMED auxin efflux carriers within the plasma membrane. Defects in defined post-Golgi trafficking processes proved causal for this phenotype and might be due to lack of direct 14-3-3 interactions with factors crucial for membrane trafficking. Taken together, our data demonstrate a fundamental role for the ancient 14-3-3 epsilon group members in regulating PIN polarity and plant development.

**\*For correspondence:** claudia. oecking@zmbp.uni-tuebingen.de

**Competing interests:** The authors declare that no competing interests exist.

## Introduction

Members of the eukaryotic 14-3-3 family are highly conserved eukaryotic proteins that have been implicated in the regulation of distinct cellular processes by protein-protein interactions. Notably, 14-3-3 proteins bind to phosphoserine/phosphothreonine motifs in a sequence-specific manner and are required to change the activity state of the respective target protein. They enforce conformational changes, act as intermolecular bridge or modify the subcellular localization of their clients (*Mackintosh, 2004*). Plant 14-3-3s have been demonstrated to be of utmost importance for the regulation of enzymes that are crucial for nutrient uptake and processing, such as the plasma membrane (PM) H$^+$-ATPase and nitrate reductase. Yet, another key aspect of 14-3-3 activity is the modification of transcriptional regulators involved in phytohormone action, such as BRASSINAZOLE-RESISTANT1, or in developmental signaling, such as the floral mediator (*de Boer et al., 2013*). While the individual studies focusing on particular 14-3-3 client proteins have generated substantial insight into the function of plant 14-3-3s, the question of functional diversity among 14-3-3 isoforms is still not fully resolved. *Arabidopsis thaliana* expresses thirteen 14-3-3 isoforms which can be divided into two major phylogenetic groups, the non-epsilon group (eight members) and the ancestral epsilon group (five members) (*Figure 1A*). Members of both groups are present in all plant genomes sequenced so far. The phylogenetic relationship of 14-3-3 isoforms from six plant species (*A. thaliana*, *Solanum lycopersicum*, *Medicago truncatula*, *Populus trichocarpa*, *Oryza sativa*, *Physcomitrella patens*) and their expression patterns (based on publicly accessible RNA-seq or microarray data) are depicted in *Figure 1—figure supplement 1* and *Figure 1—source data 1*, respectively. Except for the moss, *P. patens*, the total transcript level of all non-epsilon members in a given plant species is generally

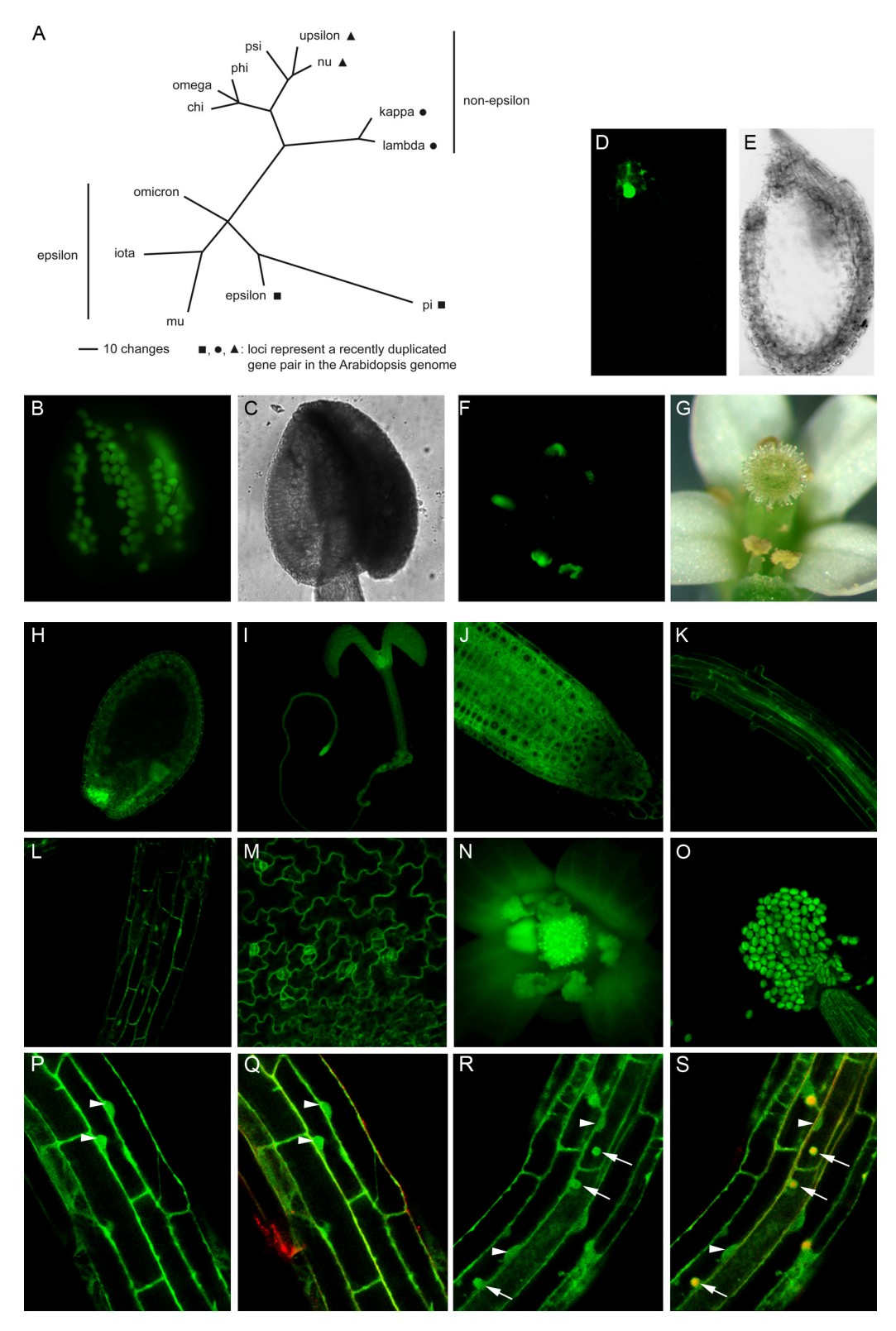

**Figure 1.** Phylogenetic tree of the *Arabidopsis* 14-3-3 isoforms and expression of genomic 14-3-3:GFP fusions under the control of the endogenous promoter. (**A**) Unrooted phylogenetic tree of the *Arabidopsis* 14-3-3 protein family. Maximum parsimony analyses were performed using PAUP 4.0b10 (Altivec) with the bootstrap-algorithm and 1000 replica. (**B, C**) Expression of iota-GFP is restricted to pollen. (**D**) to (**G**) Pi-GFP is exclusively expressed in the chalazal cyst of the seed (**D, E**) and the connective tissue of the anthers (**F, G**). (**H**) to (**O**) The expression of nu-GFP is ubiquitous. Expression in

*Figure 1 continued*

seeds (**H**), seedlings (**I**), root tips (**J**), roots (**K**), hypocotyl (**L**), leaf epidermis (**M**) and floral organs (**N**, **O**) is shown. (**P**) to (**S**) Mu-GFP (**P**, **R**) and merge with FM4-64 (**Q**, **S**) in epidermal root cells (elongation zone) of seedlings (4d) treated with CHX (50 µM) for 60 min followed by treatment with CHX (**P**, **Q**) or CHX and 50 µM BFA (**R**, **S**) for 60 min. Note that mu-GFP localizes to both nucleus and cytoplasm but accumulates in BFA bodies upon inhibitor treatment. Arrowheads indicate nuclei, arrows mark BFA bodies.

The following source data and figure supplements are available for figure 1:

**Source data 1.** Expression patterns of 14-3-3 isoforms in various tissues of six plant species based on publicly accessible RNA-seq or microarray data.

**Figure supplement 1.** Phylogenetic relationship of 14-3-3 family members from six plant species.

**Figure supplement 2.** Phenotype of the homozygous *14-3-3 mu* T-DNA allele (SALK_004455) under continuous light.

higher than that of the epsilon group isoforms. However, whether this relates to lower amounts of 14-3-3 epsilon proteins in individual cells or tissues remains unclear. The ancestral epsilon group may mediate fundamental cellular features while non-epsilon members may perform organism-specific functions (*Ferl et al., 2002*). In this regard, it has recently been shown that the *Arabidopsis* 14-3-3 isoform kappa (non-epsilon) is required for PHOTOTROPIN2-mediated blue-light-induced stomatal opening (*Tseng et al., 2012*). However, with the exception of upsilon, the knockout mutant of which flowers late under long day conditions (*Mayfield et al., 2012*), several single and double T-DNA-induced *loss-of-function* alleles of non-epsilon isoforms are indistinguishable from wild type under normal growth conditions and even quadruple mutants showed only a mild reduction of primary root length (*van Kleeff et al., 2014*). Nevertheless, under given abiotic stress conditions, a certain specificity of defined double mutants was uncovered (*van Kleeff et al., 2014*). Taken together, functional overlap seems to be a predominant issue among non-epsilon 14-3-3 members. So, what about the ancient epsilon group in *Arabidopsis*?

With the exception of omicron, no single T-DNA-induced knockout mutant of epsilon group members has been described so far. Omicron seems to participate in the regulation of iron acquisition (*Yang et al., 2013*), while a knockdown allele of mu exhibited exaggerated Pi starvation responses (*Cao et al., 2007*). The same mu allele was later shown to have shorter roots under constant light (*Mayfield et al., 2012*) or water stress (*He et al., 2015*). In summary, the in vivo role of the epsilon group is virtually unknown. Furthermore, the important question of whether the ancestral epsilon group members indeed fulfill rather basal cellular functions remains largely unresolved. Here, we tackle this question by conditional knockdown of the entire 14-3-3 epsilon group in *Arabidopsis* seedlings (three of the five isoforms are expressed in seedlings). Interference with their function led to pronounced defects in PIN polar distribution and associated auxin-mediated plant development. At the cellular level, in vivo visualization of endocytic cargo revealed defects in defined post-Golgi trafficking processes. 14-3-3 proteins co-immunoprecipitated with several factors crucial for membrane trafficking, suggesting that lack of these interactions might underlie the mutant phenotype.

## Results and discussion

### Expression of 14-3-3 isoforms

To determine the expression pattern of *Arabidopsis* 14-3-3 isoforms (*Figure 1A*), translational fusions between their genomic sequences, comprising the respective promoter, and the reporter green fluorescent protein (GFP) were constructed. The analysis of transgenic *Arabidopsis* revealed a strict tissue-specific expression pattern for two of the epsilon-group members. Expression of iota (AT1G26480) is restricted to pollen, while pi (AT1G78220) is exclusively expressed in anthers and the chalazal cyst of the seed (*Figure 1B–G*). Yet another epsilon group member, omicron (AT1G34760), is expressed in anthers too, but also moderately in roots and cotyledons.

Overall, however, the expression of the remaining 10 non-epsilon and epsilon isoforms occurs in neither a tissue-specific nor a developmentally regulated manner (isoform nu shown in *Figure 1H–O*). Thus, in *Arabidopsis*, differences in the expression patterns of 14-3-3 isoforms are not assumed to

contribute to possible isoform specificity. The same applies to the majority of isoforms expressed in *S. lycopersicum*, *M. truncatula*, *P. trichocarpa*, *O. sativa* and *P. patens* (*Figure 1—source data 1*).

## RNAi and amiRNA of 14-3-3 epsilon members interferes with plant growth and development

Unfortunately, we did not identify T-DNA-induced null alleles for most of the epsilon group members. It has previously been reported that a knockdown of the 14-3-3 isoform mu (SALK_004455) caused significantly shorter roots in constant light (*Mayfield et al., 2012*). The homozygous SALK line, however, did not show a striking phenotype in our hands (*Figure 1—figure supplement 2*; see also: [*Cao et al., 2007*; *He et al., 2015*]). Thus, we designed ethanol-inducible RNA interference (RNAi) (*Figure 2*, *Figure 2—figure supplement 1*) and artificial microRNA (amiRNA) (*Figure 2—figure supplement 1*) constructs to suppress the function of those epsilon group members that are rather ubiquitously expressed, namely epsilon (AT1G22300), mu (AT2G42590) and omicron (AT1G34760). We analyzed independent homozygous transgenic lines that are referred to as *emo*-RNAi (three lines) and amiRNA-(*em*)*o* (one line) plants, respectively.

Semiquantitative RT-PCR confirmed that the expression of the targeted 14-3-3 isoforms was efficiently reduced upon ethanol treatment (*Figure 2G*, *Figure 2—figure supplement 1C*). Notably, transcript levels of all non-epsilon isoforms were unaffected (kappa, chi, upsilon shown in *Figure 2G*). Considering that 14-3-3 isoforms are highly homologous, this clearly suggests that the risk of potential off-targets of produced siRNA is marginal.

Under inductive conditions (0.1%(v/v) ethanol), *emo*-RNAi (*Figure 2B*) and amiRNA-(*em*)*o* (*Figure 2—figure supplement 1B*) seedlings were strongly defective in growth, with highly impaired root elongation and nonexpanded, epinastic cotyledons. This is in striking contrast to the mild phenotypes reported even for quadruple knockout mutants of 14-3-3 non-epsilon isoforms (*van Kleeff et al., 2014*) and might highlight a particular importance of the ancestral epsilon group.

Etiolated *emo*-RNAi seedlings germinated on inductive medium showed a severe defect in elongation of both roots and hypocotyls (*Figure 2D*). Ethylene overproduction is not the underlying cause since the biosynthesis inhibitor aminovinylglycine (AVG) does not rescue the phenotype (*Figure 2H*). In addition, the hypocotyls are hookless even in the presence of the ethylene biosynthesis precursor 1-aminocyclopropane-1-carboxylate (ACC) (*Figure 2F*). Both apical hook development and ethylene-induced hook exaggeration have been most extensively connected with auxin (*Stepanova et al., 2008*; *Vandenbussche et al., 2010*; *Zádníková et al., 2010*). Taking into account that light grown *emo*-RNAi/amiRNA(*em*)*o* seedlings exhibit a wavy root phenotype and do not form lateral roots, the phenotypic features might be related to alterations in auxin-mediated processes. Since the independent *emo*-RNAi/amiRNA-(*em*)*o*-lines reacted in a qualitatively and quantitatively comparable manner, we focused on the characterization of *emo1*-RNAi.

## RNAi of 14-3-3 epsilon members interferes with auxin transport-dependent processes

Auxin-induced gene expression was not severely affected in roots of the mutant following ethanol treatment (*Figure 2—figure supplement 2*), indicating that auxin perception and the subsequent degradation of Aux/IAA repressors were not compromised. We therefore tested whether interference with 14-3-3 function impacts auxin-transport dependent processes, such as lateral root formation and response to gravity. Upon ethanol treatment (3 days) in the presence of the transportable synthetic auxin naphthalene acetic acid (NAA) (48 hr), discrete lateral root primordia formed along the main root of control seedlings (*Figure 3B*). By contrast, *emo1*-RNAi seedlings displayed highly disorganized proliferation of pericycle cells (*Figure 3E*). This resembles wild-type roots treated with the non-transportable auxin analogue 2,4-dichlorphenoxyacetic acid (2,4D) (*Figure 3C*) (*Geldner et al., 2004*). In addition, *emo1*-RNAi roots failed to orient their growth with respect to the gravity vector (*Figure 2—figure supplement 3A–D*) and moreover, etiolated seedlings had agravitropic hypocotyls (*Figure 2—figure supplement 3F,G*). Yet, another process that depends on polar auxin transport in the shoot is the formation of adventitious roots as a consequence of surgical removal of the primary root (*Geldner et al., 2004*). As expected, wildtype seedlings spontaneously regenerated a root from the hypocotyl stump, while *emo1*-RNAi seedlings were unable to form adventitious roots (*Figure 2—figure supplement 3M–P*). Together, these data show that the

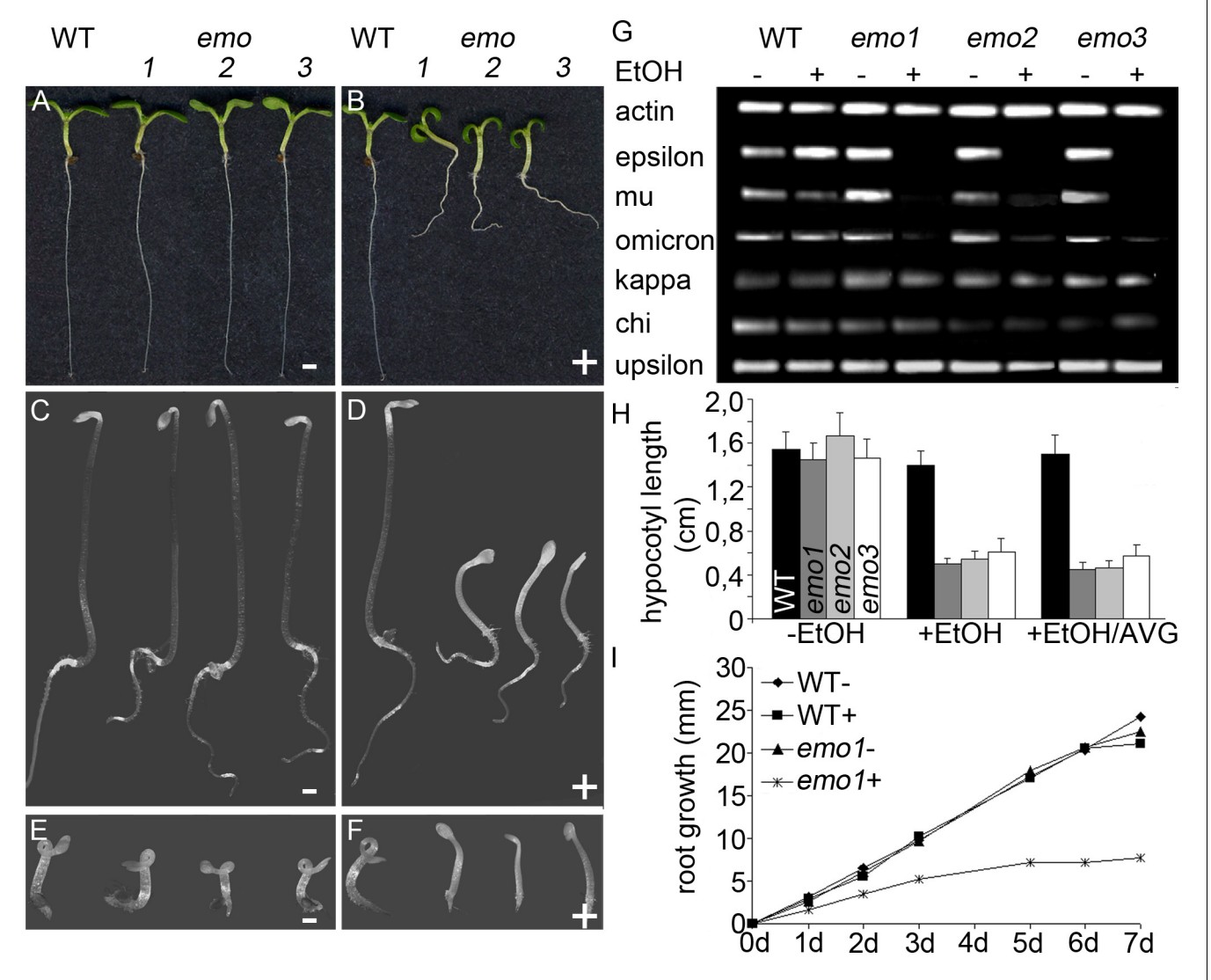

**Figure 2.** Ethanol-inducible *emo*-RNAi causes growth retardation phenotypes. (**A**) to (**F**) Seedlings (wildtype and three independent *emo*-RNAi lines) grown either for 6 days in the light (**A, B**) or for 4 days in the dark (**C–F**) on noninductive (**A, C, E**) or inductive (0.1% (v/v) EtOH) medium (**B, D, F**), optionally supplemented with ACC (10 µM) (**E, F**). (**G**) Semiquantitative RT-PCR analysis of the transcript level of selected 14-3-3 isoforms in light-grown seedlings. (**H**) Hypocotyl length of etiolated seedlings (n = 30). (**I**) Primary root growth of seedlings grown for 4 days in the absence of ethanol followed by transfer to non-inductive (−) or inductive (+) medium for the indicated times (n = 16).

The following figure supplements are available for figure 2:

**Figure supplement 1.** Ethanol-inducible amiRNA-(*em*)o causes growth retardation phenotypes.

**Figure supplement 2.** Auxin-induced gene expression is not compromised in *emo*-RNAi roots.

**Figure supplement 3.** Ethanol-inducible *emo*-RNAi causes defects in the gravitropic growth response and auxin transport in both roots and aerial tissues.

function of 14-3-3 epsilon group members is required for auxin-transport-dependent development in both roots and shoots.

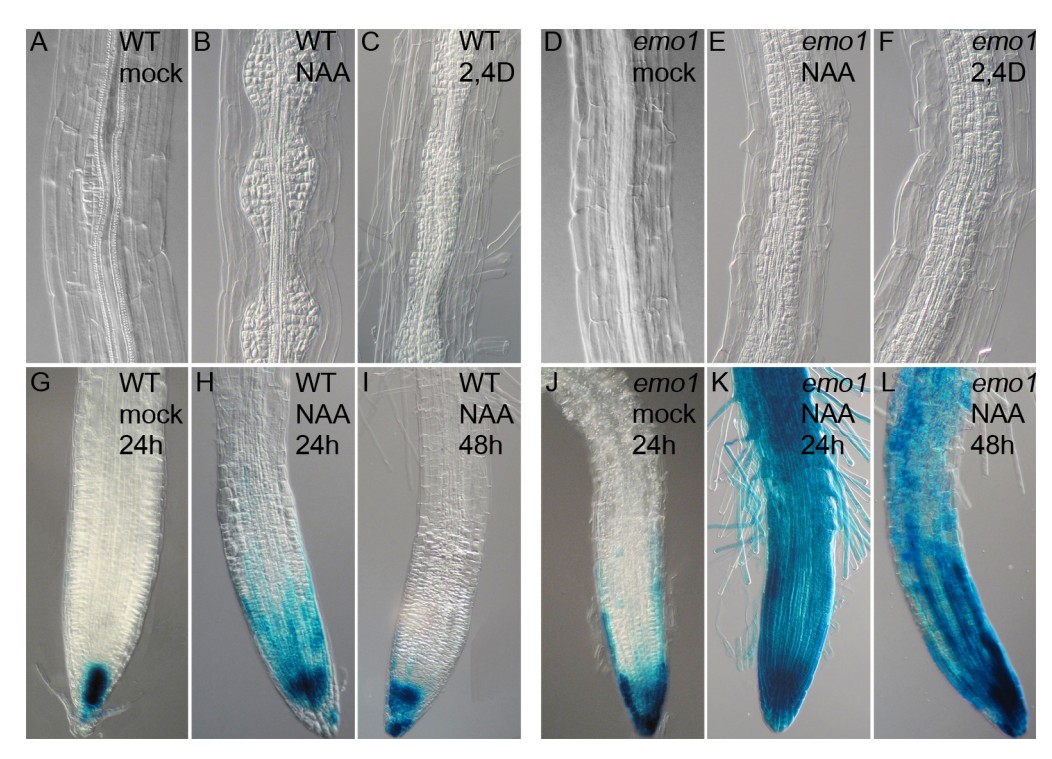

**Figure 3.** Disorganized lateral root primordia and failure in the establishment of auxin response gradients caused by *emo*-RNAi induction. Seedlings grown for 4 days in the absence of ethanol were transferred to inductive medium for 24 hr followed by treatment with exogenous auxin.(**A**) to (**F**) Lateral root primordia of wildtype (**A–C**) and *emo1*-RNAi (**D–F**) after 48 hr of treatment with either mock solution (**A, D**), 1 μM NAA (**B, E**) or 1 μM 2,4D (**C, F**) in the presence of ethanol. (**G**) to (**L**) Auxin response gradients visualized by *DR5::GUS* in wild type (**G–I**) and *emo1*-RNAi (**J–L**) after treatment with either mock solution (**G, J**) or 0.1 μM NAA for either 24 hr (**H, K**) or 48 hr (**I, L**) in the presence of ethanol.

The following figure supplement is available for figure 3:

**Figure supplement 1.** Changes in auxin distribution caused by *emo*-RNAi induction.

## RNAi of 14-3-3 epsilon members interferes with auxin distribution

Because gravitropic growth and formation of both lateral and adventitious roots depend on dynamic auxin redistribution (*Geldner et al., 2004*; *Luschnig et al., 1998*; *Benková et al., 2003*), we investigated the effect of RNAi induction on auxin distribution with the auxin reporter *DR5::GUS* (*Ulmasov et al., 1997*; *Sabatini et al., 1999*) as well as *DR5rev::GFP* (*Friml et al., 2003*).

In wild-type root meristems, DR5 was highly active in the columella and in stele tissues (*Figure 3G*, *Figure 3—figure supplement 1A*). By contrast, induction of *emo1*-RNAi caused an ectopic accumulation of GUS activity (*Figure 3J*) / expression of GFP (*Figure 3—figure supplement 1B*) in the entire lateral root cap. As reported previously, roots have homeostatic mechanisms that maintain the endogenous auxin gradients in the presence of exogenously applied auxin (*Geldner et al., 2004*; *Friml et al., 2002*). Exposure to low concentrations of the transportable auxin NAA did hence not lead to strong alterations in DR5::GUS staining in control roots (*Figure 3H,I*), whereas GUS activity expanded across the entire root of *emo1*-RNAi seedlings (*Figure 3K,L*). Apparently, *emo* roots have a reduced capacity to maintain auxin gradients in the presence of exogenous auxin.

After gravistimulation of the wild type, the DR5*rev*::GFP signal relocated asymmetrically to the gravistimulated side (*Figure 3—figure supplement 1E,G*), indicating a shift in auxin distribution which is essential for the differential growth response (*Ottenschläger et al., 2003*). By contrast, the

ectopic lateral expression pattern of DR5rev::GFP in the root meristem of *emo1*-RNAi persisted, independent of gravistimulation (*Figure 3—figure supplement 1F,G*). Taken together, these data suggest that interference with 14-3-3 epsilon group members affects auxin distribution.

Therefore, the effect of reduced 14-3-3 expression on auxin transport toward the root apex was analyzed. As compared with the non-induced *emo1*-RNAi control, auxin transport to the root tip decreased upon ethanol induction (*Figure 2—figure supplements 3E*, 24 hr: 22%, 48 hr: 42%, n = 10) and, furthermore, the gradual decrease in auxin response gradients within the root tip could be visualized by using the *DR5rev::GFP* reporter (*Figure 3—figure supplement 1B–D*).

However, another striking aspect of the RNAi-mutant upon transfer to ethanol-containing medium is an epinasty cotyledon phenotype (*Figure 2B*), characteristic of auxin overproducers (*Zhao et al., 2001*; *Boerjan et al., 1995*). Consequently, in contrast to the situation in roots, the auxin level might be elevated in *emo* hypocotyls and cotyledons. Accordingly, the mutant showed a severe increase in *DR5::GUS* reporter activity in the cotyledons upon induction (*Figure 2—figure supplement 3L*). Intriguingly, treatment of the wild type with the auxin transport inhibitor NPA resulted in agravitropic hypocotyls (*Figure 2—figure supplement 3H*), an increase in GUS activity in cotyledons (*Figure 2—figure supplement 3J*) and the formation of epinastic leaves (*Geldner et al., 2004*). Taken together, this strongly suggests that suppression of 14-3-3 expression impairs auxin transport in both roots and aerial tissues, resulting in depletion of auxin in the root tip and an elevated auxin level in the apical part.

## 14-3-3 epsilon group members: master regulators of the H$^+$-ATPase?

The phenotypic analysis of *emo* mutants revealed an impressive number of auxin-related, in particular auxin transport-related, phenotypes. This, however, does not necessarily imply that 14-3-3 members regulate auxin transport directly. In fact, several mutants affected in rather fundamental eukaryotic processes, such as vesicular trafficking events, exhibit auxin-related phenotypes (*Geldner et al., 2003*; *Jaillais et al., 2006*; *Kitakura et al., 2011*; *Mravec et al., 2011*). What might be the major 14-3-3 regulated cellular process or 14-3-3 target protein(s) causing the phenotype? According to the chemiosmotic hypothesis for polar auxin transport, the activity of the PM localized H$^+$-ATPase is crucial for auxin uptake into the cell and, consequently, directed cellular export mediated by auxin efflux facilitators. Regulation of this P-type proton pump is one of the most highly explored functions of plant 14-3-3 proteins. In brief, phosphorylation-dependent association of 14-3-3 with the penultimate residue results in a displacement of the C-terminal autoinhibitor and hence activation of the pump (*Speth et al., 2010*). Dexamethasone (Dex)-dependent expression of one of the major H$^+$-ATPase isoforms, AHA2, deleted of its C-terminal autoinhibitor (94 amino acids, 1-887, AHA2$^{\Delta 94}$) (*Pacheco-Villalobos et al., 2016*) and thus representing a constitutively active version of the H$^+$-pump uncoupled from regulatory inputs, did not rescue the *emo1*-RNAi phenotype (*Figure 4A–E*). Notably, the effects observed in wild type upon Dex treatment (shorter hypocotyls, absence of an apical hook and open cotyledons) are indeed the consequence of an activation of the H$^+$-ATPase since they can be mimicked by a specific activator of the H$^+$-pump, fusicoccin (*Würtele et al., 2003*) (*Figure 4F,G*). Therefore, inhibition of the H$^+$-ATPase is not the underlying cause of the *emo1*-RNAi phenotype.

Another thought comes to mind: might 14-3-3 proteins interact directly with auxin efflux carriers such as PIN proteins? The PIN transporters are characterized by a large hydrophilic loop localized in the cytoplasm and phosphorylated at multiple sites (*Huang et al., 2010*; *Dhonukshe et al., 2010*; *Zourelidou et al., 2014*). Interaction of 14-3-3 epsilon and the PIN2 hydrophilic loop (amino acids 187-477) was not detectable by yeast two-hybrid assay and ratiometric bimolecular fluorescence complementation *in planta* (*Grefen and Blatt, 2012*) (*Figure 4—figure supplement 1*). This, however, is no sufficient proof for the lack of direct association of 14-3-3 with PIN proteins.

## RNAi of 14-3-3 epsilon members interferes with PIN expression and polar localization

Given that RNAi of the 14-3-3 epsilon group members interferes with auxin transport, we analyzed expression and localization of members of the PIN familiy of auxin efflux facilitators in roots.

Under our experimental conditions, PIN1-GFP (*pPIN1:PIN1-GFP*, [*Benková et al., 2003*]) was detected predominantly at the basal side of stele and endodermis cells of the wild type with

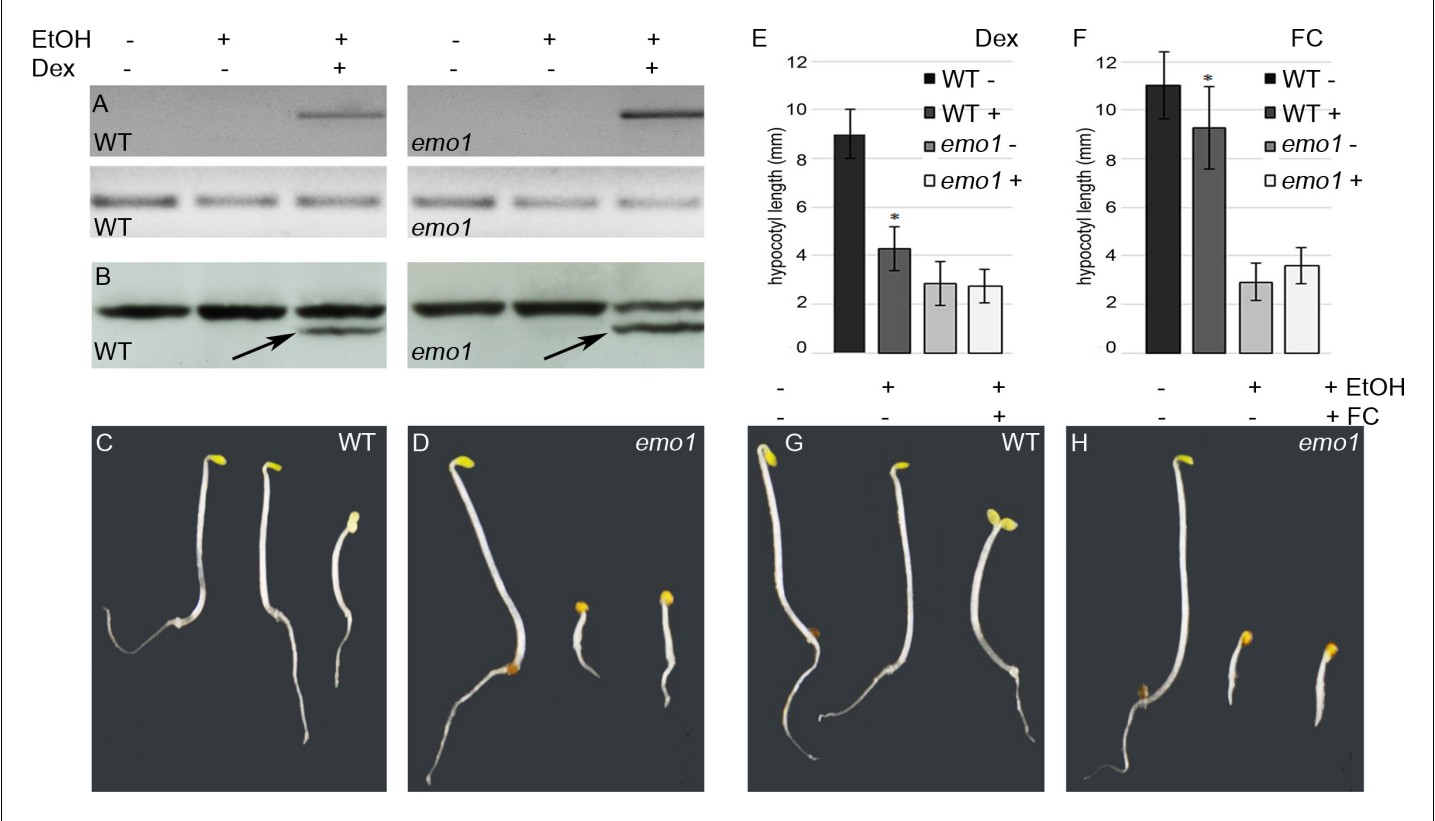

**Figure 4.** Dexamethasone-dependent expression of a constitutively active version of AHA2 does not rescue the *emo*-RNAi phenotype. (**A**) to (**E**) Wild-type and *emo1*-RNAi seedlings were grown for 4 days in the dark in the absence or presence of ethanol and Dex (10 μM). Semiquantitative RT-PCR analysis of the transcript level of the C-terminally deleted AHA2 (**A**, upper panel, lower panel shows actin) and immunodetection of the H⁺-ATPase in microsomal membranes (**B**, arrows indicate the truncated version of the H⁺-pump) confirmed Dex-dependent expression of the transgene. Seedling growth (**C, D**) and measurement of the hypocotyl length (**E**, n = 30) is shown. (**F**) to (**H**) Treatment with fusicoccin (FC), an activator of the H⁺-ATPase, mimicked the effects observed in transgenic wild type upon Dex-treatment (see (**C**): shorter hypocotyls, absence of an apical hook and open cotyledons). Seedling growth (**G, H**) and measurement of the hypocotyl length (**F**, n = 30) in the absence or presence of FC is shown.

The following figure supplement is available for figure 4:

**Figure supplement 1.** Yeast two-hybrid analysis and *in planta* interaction studies do not point to a direct interaction of 14-3-3 s with the PIN2 hydrophilic loop.

occasional weak expression in cortex cells (***Figure 5A,B***). PIN2-GFP (*pPIN2:PIN2-GFP*; [***Xu and Scheres, 2005***]) is expressed in older epidermis and cortex cells exhibiting apical and basal polarity, respectively (***Figure 5G***). Interference with 14-3-3 function resulted in misexpression of the two auxin efflux carriers. PIN1-GFP fluorescence was observed in cortex cells with an intensity comparable to that in endodermis cells and also moderately in the epidermis (***Figure 5C,D***). PIN2-GFP showed mis-expression in the younger epidermis and cortex daughter cells and in the entire lateral root cap as well (***Figure 5H***). Crossregulation of *PIN* gene expression has been described in *pin* mutants or plants with inhibited auxin transport (***Vieten et al., 2005***) and thus, the defects observed in *emo1*-RNAi roots are probably the consequence of the interference with auxin homeostasis.

Notably, the subcellular polar localization of PIN-GFP proteins was reduced as a consequence of RNAi induction. In *emo1*-RNAi, the PIN1-GFP signal was not restricted to the basal side of those cells showing misexpression. In fact, PIN1-GFP was additionally detectable in the outer lateral PM-side of endodermal cells and did not show pronounced polarity in cortex cells (***Figure 5D,E***). Moreover, by contrast with the wild type, PIN2-GFP fluorescence is frequently visible in the inner lateral PM-side of cortex cells upon interference with 14-3-3 function (***Figure 5F,H***).

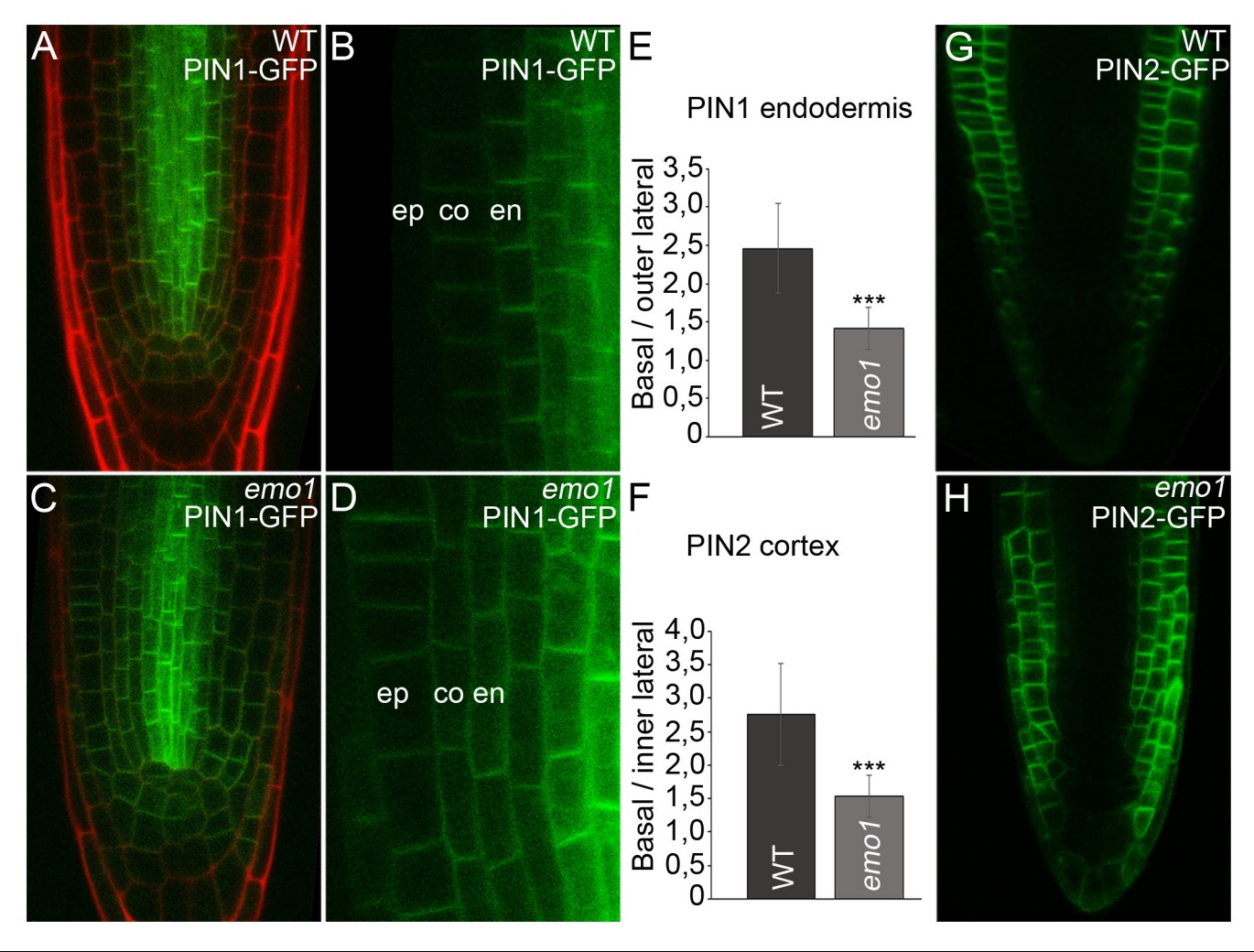

**Figure 5.** Ethanol-inducible *emo*-RNAi causes misexpression of PIN proteins in root tips. (A) to (H) Expression of PIN1-GFP (A–D) and PIN2-GFP (G, H) in wildtype (A, B, G) and *emo1*-RNAi (C, D, H) seedlings grown for 4 days on inductive medium. The ratio of GFP intensity on the basal to that on the lateral side of the plasma membrane is shown (E, n ≥ 64 cells, 12 roots, F, n ≥ 75 cells, 10 roots). en: endodermis, co: cortex, ep: epidermis.

## RNAi of 14-3-3 epsilon members interferes with endocytic membrane trafficking

PIN proteins continuously cycle between endosomes and the PM via vesicle trafficking allowing for the establishment and rapid alteration of polarity (*Geldner et al., 2004*; *Dhonukshe et al., 2007*). Furthermore, PIN proteins are sorted to the lytic vacuole via late endosomes (LE)/multivesicular bodies (MVB), resulting in protein degradation and modulation of auxin fluxes (*Kleine-Vehn et al., 2008*). Accordingly, treatment with vesicle trafficking inhibitors affects PIN trafficking/localization: Brefeldin A (BFA) causes PIN accumulation in intracellular, so-called BFA compartments encompassing aggregation of early endosomes (EE)/trans-Golgi network (TGN) and recycling endosomes that get surrounded by MVBs and the golgi apparatus (*Robinson et al., 2008*). We noticed that the BFA-induced PIN2-GFP accumulation was more pronounced in ethanol-treated root epidermis cells of *emo1*-RNAi as compared to the wild type (*Figure 6A–F,K*), irrespective of the presence of the protein synthesis inhibitor cycloheximide (CHX) (*Figure 6G,I*). A comparable scenario has been described for *ric1-1* (*Chen et al., 2012*) and *bex5-1* (*Feraru et al., 2012*) mutants, which are affected in endocytosis and PIN recycling to the plasma membrane, respectively. Endocytosis, however, does not seem to be significantly altered in *emo1*-RNAi as evidenced by short-term uptake of the

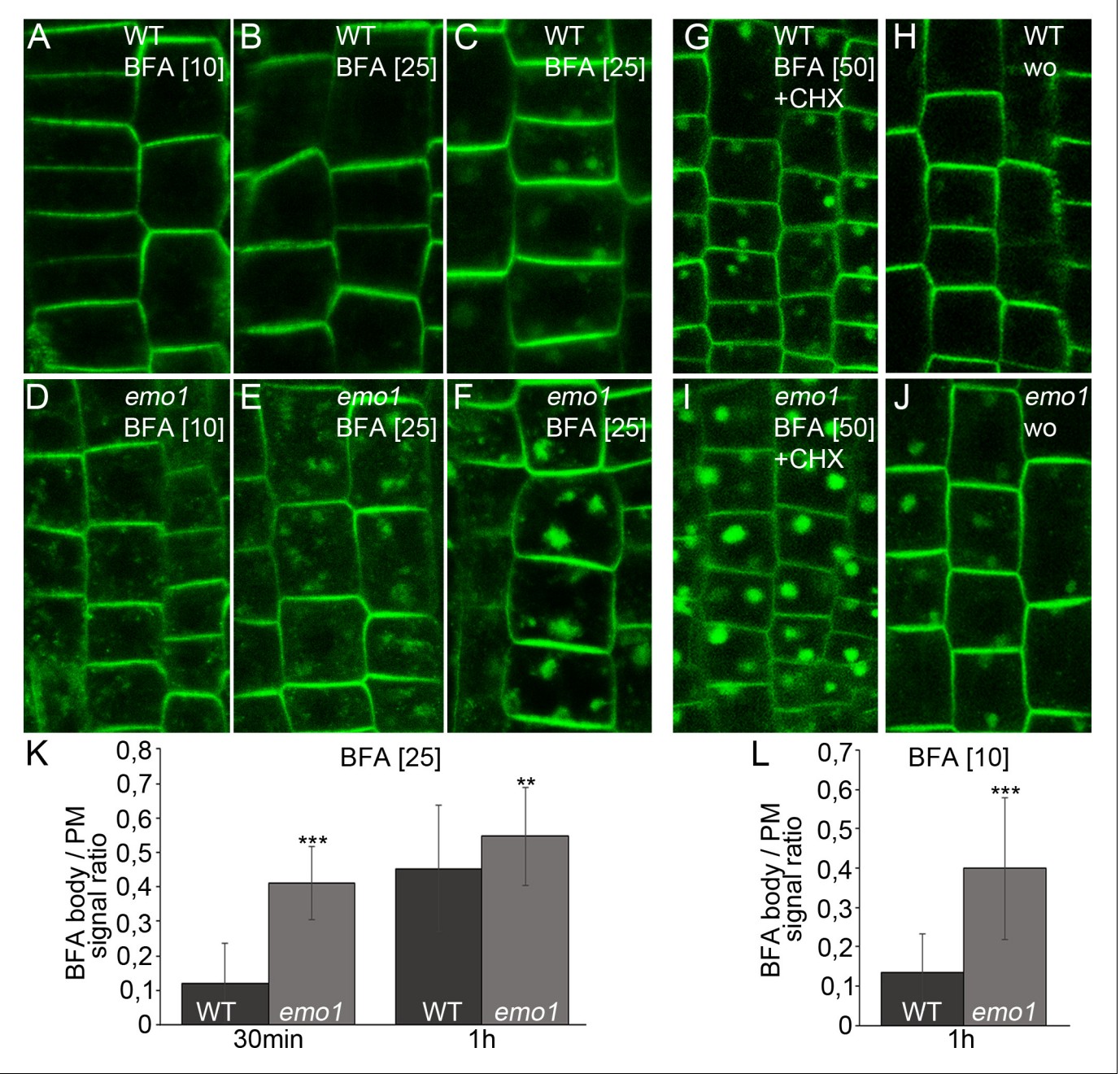

**Figure 6.** Ethanol-inducible *emo*-RNAi alters membrane trafficking processes. (A) to (F) PIN2-GFP in epidermal root cells of wild-type (A–C) and *emo1*-RNAi (D–F) seedlings grown for 4 days on inductive medium and treated with BFA for 10 min (A, D), 30 min (B, E) or 60 min (C, F). The BFA concentration used (µM) is given in brackets. (G) to (J) PIN2-GFP in epidermal root cells of induced (4d) wild-type (G, H) and *emo1*-RNAi (I, J) seedlings treated with cycloheximide (CHX, 50 µM) for 60 min followed by concomitant treatment with CHX and 50 µM BFA (G, I) and subsequent washout of BFA (H, J) for 60 min (wo), respectively. (K, L) Ratio of PIN2-GFP intensity in BFA bodies and the PM (1 BFA body and 1 PM per cell, K: n ≥ 90 cells, 15 roots; L: n ≥ 75 cells, 11 roots).

endocytic tracer FM4-64 (*Figure 7—figure supplement 1A,D*). Strikingly, the mutant displayed BFA hypersensitivity even when treated with 10 µM BFA (*Figure 6L*), a concentration which preferentially inhibits recycling and minimizes the interference with vacuolar trafficking (*Robert et al., 2010*; *Chen et al., 2012*). This suggests that exocytosis might be affected in *emo1*-RNAi. Therefore, a washout experiment after BFA treatment was performed. While the PIN2-containing BFA

compartments efficiently disappeared in wild-type root epidermal cells (washout efficiency: 66%, n ≥ 100 cells, 20 roots), BFA-induced PIN2-GFP agglomerations persisted in *emo1*-RNAi (washout efficiency: 39%, n ≥ 100 cells, 20 roots) (*Figure 6G–J*). Taken together, reduced expression of the 14-3-3 epsilon members interferes with PIN2 recycling events to the PM.

Next, endocytic transport to the vacuole was analyzed. The Rab5-like GTPase ARA7 marks a subdomain of the TGN but mostly accumulates at the MVBs (*Singh et al., 2014*) that become vacuolated in the presence of wortmannin (*Robinson et al., 2008*). While striking differences could not be observed in the absence of inhibitor, the size of ARA7-containing wortmannin-induced compartments displaying ring-shaped structures was remarkably reduced in *emo1*-RNAi as compared to the wild type (*Figure 7A–G*). Such a phenotype has – to our knowledge – not been described so far. Enlargement of ARA7-positive organelles is, however, characteristic of mutants such as *gnom* (impaired recycling: [*Geldner et al., 2003*]), *vps29* (impaired retrograde trafficking: [*Jaillais et al., 2007*]) and *sand* (impaired MVB-vacuole fusion: [*Singh et al., 2014*]), which are affected in diverse post-Golgi trafficking processes but all interfere (indirectly or directly) with vacuolar targeting. Accordingly, we revealed through pulse labeling by FM4-64 and sequential treatments of cycloheximide and BFA that endosomal trafficking to the tonoplast membrane is significantly delayed in *emo1*-RNAi in comparison to the wild type (*Figure 7H–N*). The retarded trafficking of FM4-64 and thus, of PM material in general, to the vacuole is also evident in the absence of BFA (*Figure 7—figure supplement 1B,E*). Consistent with the observed effects on vacuolar trafficking, *emo1*-RNAi exhibits an increase in the total PIN2-GFP levels in membrane protein preparations in the presence of ethanol (*Figure 7—figure supplement 1H*), while transcript levels remain unaltered.

At the cellular level, the absence of 14-3-3 epsilon group members did hence entail disturbance of at least two endocytic trafficking pathways: from the TGN to the vacuole and to the PM. We thus assumed the level of PIN2-GFP to be elevated in intracellular vesicular structures of *emo1*-RNAi. Indeed, in contrast to the wild type, considerable levels of PIN2-GFP are detectable in endosomal structures in *emo1*-RNAi following short-term treatment (10 min) with BFA concentrations as low as 10 µM (*Figure 6A,D*). Auxins such as NAA inhibit endocytosis but do not interfere with the BFA-induced formation of endosomal BFA compartments (*Paciorek et al., 2005*). When pretreated with FM4-64 (5 min), we observed a reduced frequency and size of FM-stained BFA bodies in NAA (and CHX)-treated plants. Nonetheless, in contrast to the wild type, PIN2-GFP localized to BFA compartments of *emo1*-RNAi root epidermal cells (*Figure 7—figure supplement 1I–M*), strongly supporting an increased PIN2-GFP content in endosomal structures. Taken together, our finding clearly points to an accumulation of PIN2-GFP in endosomal vesicles such as early/recycling endosomes and MVBs.

## 14-3-3 epsilon group members may directly regulate trafficking processes

Since trafficking processes were affected by the RNAi approach, we wondered whether the respective 14-3-3 isoforms localize to post Golgi compartments. Indeed, they do since BFA clearly induced the accumulation of epsilon group members in BFA bodies (mu: *Figure 1R,S*).

Next, we identified potential targets of 14-3-3 epsilon-GFP by stringent co-immunoprecipitation experiments coupled with MS-based protein identification (*Table 1—source data 1*). As expected, several well-characterized 14-3-3 interactors were identified by MS (highlighted in green in *Table 1—source data 1*), among those different isoforms of the plasma membrane H+-ATPase. Furthermore, a remarkable number of proteins known or assumed to be involved in membrane trafficking processes co-precipitated with epsilon-GFP but not GFP alone (*Table 1*). These include small GTPases (subfamily RAB) and their effector proteins (subfamilies ARF and RAB), required for vesicle formation (ARF) and tethering (RAB) (*Park and Jürgens, 2011*; *Vernoud et al., 2003*). Likewise, AtTRS130 has been suggested to act upstream of RAB-A GTPases in post-Golgi membrane trafficking (*Qi et al., 2011*). In addition, adaptors of membrane vesicle coat proteins involved in clathrin-related endomembrane trafficking in plants (ENTH/ANTH/VHS superfamily proteins) (*Zouhar and Sauer, 2014*), proteins essential for vesicle fusion, such as regulators of SNAREs (e.g. SEC1B) and a regulator of the MVB pathway (IST1-LIKE 3) were identified as putative epsilon interactors (*Table 1*). The list of potential epsilon interactors thus strongly suggests a direct regulation of cellular trafficking by 14-3-3 proteins.

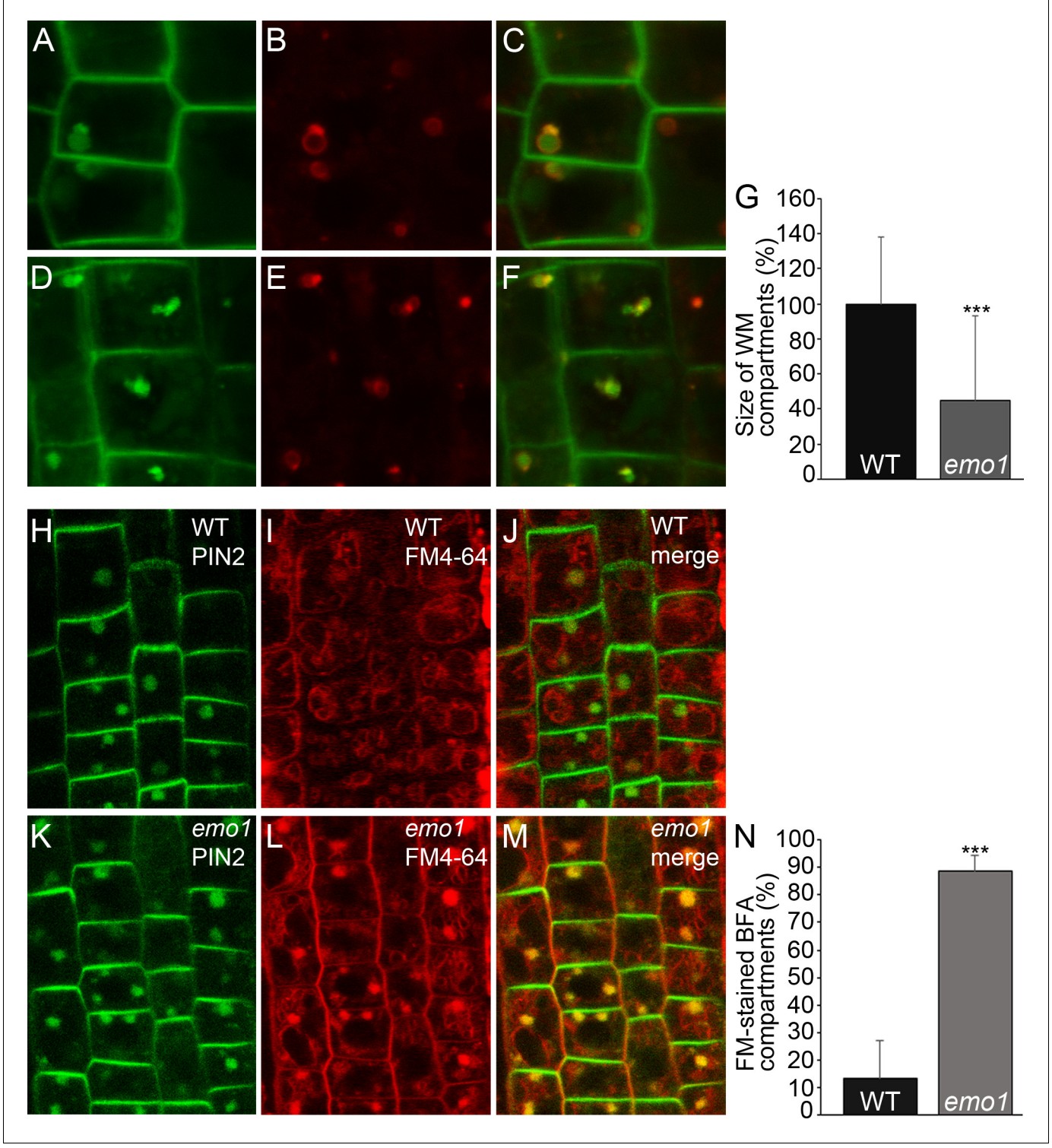

**Figure 7.** Ethanol-inducible *emo*-RNAi causes defects in wortmannin-compartment formation and endocytic transport to the vacuole. (**A**) to (**G**) PIN2-GFP (**A, D**) and mRFP-ARA7 (**B, E**) in epidermal root cells of induced (4d) wildtype (**A–C**) and *emo1*-RNAi (**D–F**) seedlings treated with cycloheximide (CHX, 50 μM) for 60 min followed by concomitant treatment with CHX and WM (15 μM) for 120 min. Measurement of the relative size of WM-compartments is shown in (**G**) (n = 100). (**H**) to (**N**) PIN2-GFP (**H, K**) and FM4-64 (**I, L**) in epidermal root cells of induced (4d) wild-type (**H–J**) and *emo1*-RNAi (**K–M**) seedlings pulse labeled by FM4-64 (5 min) and subsequently treated with CHX (50 μM, 1 hr) followed by concomitant treatment with CHX and 50 μM BFA for 1 hr. Percentage of FM4-64 stained BFA bodies (**J, M**) is shown in (**N**) (n ≥ 115 cells, 10 roots).

*Figure 7 continued on next page*

*Figure 7 continued*

The following figure supplement is available for figure 7:

**Figure supplement 1.** Ethanol-inducible *emo* RNAi delays endocytic trafficking from TGN/EE to the vacuole and alters endosome homeostasis.

## Conclusions

Our data provide insight into functions of 14-3-3 epsilon members in auxin transport-dependent plant growth and development. We applied ethanol-inducible RNAi as well as amiRNA to simultaneously reduce the expression of three epsilon group members (epsilon, mu, omicron: *emo*-RNAi and amiRNA-(*em*)*o*). Considering that the residual 'non-target' isoforms (iota, pi) are exclusively expressed in floral organs and seeds, such an approach results in knockdown of the entire epsilon

**Table 1.** Analysis of 14-3-3 epsilon-GFP immunoprecipitates via mass spectrometry (MS) based on two biological replicates. This table lists only proteins with a possible role in membrane trafficking. Proteins depicted in a clustered manner (without blank line) represent alternative possibilities based on the MS-identified peptides.

| AGI code | Gene name | Description |
|---|---|---|
| At1g08680 | AGD14 | ADP-ribosylation factor (ARF) GTPase-activating protein |
| At1g09630 | RAB-A2a | Member of the RAB-A subfamily of small Rab GTPases |
| At1g16920 | RAB-A1b/BEX5 | |
| At3g15060 | RAB-A1g | |
| At4g18800 | RAB-A1d | |
| At5g45750 | RAB-A1c | |
| At5g60860 | RAB-A1f | |
| At1g12360 | KEULE/SEC11 | SNARE-interacting protein Sec1 protein |
| At1g14670 | | Endomembrane protein 70 protein family |
| At2g01970 | | Endomembrane protein 70 protein family |
| At5g37310 | | |
| At2g20790 | AP5M | AP-5 complex subunit mu |
| At2g25430 | | ENTH/ANTH/VHS superfamily protein |
| At2g37550 | AGD7 | ARF GTPase-activating protein AGD7 |
| At2g43160 | EPSIN2 | ENTH/ANTH/VHS superfamily protein |
| At3g59290 | EPSIN3 | |
| At3g09900 | RAB-E1e | Member of the RAB-E subfamily of small Rab GTPases |
| At3g46060 | RAB-E1c/ARA3 | |
| At3g53610 | RAB-E1a | |
| At5g03520 | RAB-E1d | |
| At3g53710 | AGD6 | ARF GTPase-activating protein AGD6 |
| At4g12120 | SEC1B | Member of KEULE gene family |
| At4g32285 | | ENTH/ANTH/VHS superfamily protein |
| At4g35730 | IST1-LIKE 3 | Regulator of Vps4 activity in the MVB pathway protein |
| At5g52580 | | RAB GTPase activator activity |
| At5g54440 | ATTRS130 | TRAPII tethering factor, CLUB |

**Source data 1.** Complete list of 14-3-3 epsilon interactors based on two biological replicates. Proteins listed in *Table 1* are shown in bold face while well characterized 14-3-3 clients are highlighted in green.

group in seedlings. This gives rise to severe phenotypes as observed in plants treated with inhibitors of auxin transport or, alternatively, vesicle trafficking inhibitors, such as BFA or wortmannin (*Jaillais et al., 2006*; *Geldner et al., 2001*). Intriguingly, auxin distribution, polar auxin transport and the subcellular polar distribution of PIN efflux carriers are severely affected in *emo*-RNAi mutants, and defects in post-Golgi trafficking processes, in particular recycling from the TGN to the PM as well as endocytic transport to the vacuole, are indeed causative for this phenotype. A direct regulation of trafficking processes by 14-3-3s is corroborated by the facts that (i) the targeted 14-3-3 isoforms indeed localize to post-Golgi compartments and (ii) several key factors of endosomal trafficking co-precipitated with 14-3-3 epsilon (see below).

The epsilon group may have retained functional characteristics of the ancestral eukaryotic 14-3-3 protein. Potential roles of yeast 14-3-3 in vesicular transport have been proposed (*Gelperin et al., 1995*). Furthermore, proteome-wide surveys of the mammalian 14-3-3 interactome unequivocally revealed links to regulation of cellular trafficking, organization and membrane dynamics (*Pozuelo Rubio et al., 2004*; *Jin et al., 2004*). The molecular interactions and proper mechanisms are, however, still poorly understood with one prominent exception: 14-3-3 binding overcomes ER retention in yeast and mammals and thus positively affects forward transport of certain membrane proteins, in particular channels, in the secretory pathway (review: [*Mrowiec and Schwappach, 2006*]) In plants, large-scale experiments searching for putative 14-3-3 interactors have mostly been performed by using non-epsilon isoforms that predominantly impact primary metabolism (*Shin et al., 2011*), ion homeostasis and signal mediators (*Chang et al., 2009*). Therefore, regulation of intracellular trafficking events by plant 14-3-3 proteins has not even been considered. Interference with the function of *Arabidopsis* epsilon group members now clearly evidenced a reduced polar localization of PIN proteins and defects in endocytic trafficking pathways. We focused mainly on the analysis of PIN2 trafficking since auxin-related phenotypes dominated in *emo*-RNAi mutants. This, however, does not rule out that trafficking of many other PM proteins is affected. In fact, endocytic trafficking (TGN to vacuole) of PM material, in general, is delayed as shown by the experiments performed with the endocytic tracer FM4-64. The predominance of auxin transport-dependent phenotypes most likely is caused by the strong impact of auxin distribution on plant growth and development.

The molecular mechanisms regulating intracellular transport, such as vesicle formation, budding, delivery, tethering and fusion, are complex and not well understood in plants. Large protein families, among those small GTPases and their regulators, are crucial factors of vesicle trafficking pathways (*Park and Jürgens, 2011*). We identified several such family members required for vesicle formation, tethering and fusion as potential 14-3-3 interactors. The function and subcellular localization of the individual family members remain, however, largely uncharacterized to date. It is, nonetheless, obvious that lack of these interactions might be responsible for the phenotypes observed in *emo*-RNAi lines. Small GTPases and their regulators, for instance, represent ideal targets of 14-3-3 epsilon members due to their evolutionarily conserved function throughout eukaryotes. Remarkably, the clustering analysis of in vivo 14-3-3-binding proteins in mammals argues for significant control of small GTPase pathways by phospho-dependent 14-3-3 interactions (*Jin et al., 2004*). The detailed functional and regulatory analyses of the respective putative plant 14-3-3 target proteins are hence fascinating questions for future research.

In conclusion, our results provide clear evidence for the involvement of 14-3-3 epsilon members in regulating the subcellular polarity of PIN proteins, endocytic membrane trafficking processes and auxin-dependent growth in *Arabidopsis*.

## Materials and methods

### Plasmid constructs

Ethanol-inducible RNAi: Three individual fragments derived from the cDNA of either the 14-3-3 isoform epsilon (AT1G22300, CDS 387–498), mu (AT2G42590, CDS 213–358) or omicron (AT1G34760, CDS 387–498) were amplified by PCR and sequentially cloned as inverted repeats into a derivative of pHANNIBAL (*Wesley et al., 2001*) in which the 35S promoter has been replaced by the ethanol-inducible promoter pAlcA (*Roslan et al., 2001*). Subsequently, the pAlcA::

*epsilon_omicron_mu_RNAi* cassette was cloned into the *Not*I site of the plant transformation vector pBART_AlcR (*Figure 2—figure supplement 1*).

## Ethanol-inducible amiRNA

The amiRNAs (one designed to silence epsilon and mu simultaneously, the second designed to silence omicron) were PCR amplified according to the protocol provided via WMD3 Web MicroRNA Designer (http://wmd3.weigelworld.org) and individually subcloned. The two different amiRNA cassettes were then sequentially cloned into pBJ36_AlcA. Finally, the pAlcA-driven cassette containing both amiRNA constructs was cloned via the *Not*I site into pBART_AlcR.

For dexamethasone-dependent expression of AHA2 devoid of its autoinhibitory C-terminal domain (AHA2[894], aa 1–854, AT4G30190.1), the corresponding cDNA was PCR amplified and cloned via XhoI and SpeI into pTA7002 (*Aoyama and Chua, 1997*) (*Pacheco-Villalobos et al., 2016*).

A complete list of oligonucleotides used for PCR and RT-PCR is provided below.

## Plant materials and growth conditions

Seeds of *A. thaliana* wild-type (ecotype Col-0) and transgenic lines were surface sterilized and treated for 48 hr at 4°C before planting. Seedlings were grown at 20°C in continuous light (90 µmol $m^{-2}s^{-1}$) or darkness (after exposure to fluorescent white light for 4 hr) on solid MS medium (pH 5.8) containing 1% (w/v) sucrose optionally supplemented with 0.1% (v/v) ethanol. For hypocotyl or root length measurements, seedlings were sandwiched between two sheets of acetate and scanned in a flatbed scanner. The digitized images were analyzed using the NIH image software. For gravitropic analysis, 4-day-old seedlings were transferred to ethanol containing solid MS medium for 2 days followed by a gravistimulation (90° or 135° turn) for the indicated time. Independent experiments were carried out at least in triplicate with the same significant results. Representative images are presented. Statistics were evaluated with Excel (Microsoft).

## Transgenic plants

Seeds of transgenic *Arabidopsis* lines carrying *DR5rev::GFP* or *PIN1:GFP* (*Benková et al., 2003*) were obtained from the Nottingham Arabidopsis Stock Centre. *DR5::GUS* (*Ulmasov et al., 1997*), *PIN2::PIN2:GFP* (*Xu and Scheres, 2005*) and *35S::mRFP-ARA7* (*Beck et al., 2012*) transformed lines were obtained from Gerd Jürgens (ZMBP, Tübingen, Germany).

## Auxin treatment and application of drugs

For the analysis of auxin-induced gene expression and for induction of lateral root formation as well as for *DR5::GUS* staining of primary root tips, seedlings grown on plates were transferred into 24-well culture plates containing 1.5 mL of liquid medium supplemented with auxin or mock solvent and incubated for the indicated times. Stock solutions (100 mM) in DMSO were used for naphthalene acetic acid (NAA), indole-3-acetic acid (IAA) and 2,4 dichlorphenoxyacetic acid (2,4D).

Exogenous drugs were applied by incubation of 4-day-old seedlings in liquid MS medium supplemented with BFA (50 mM stock in DMSO) (10, 25 or 50 µM), wortmannin (50 mM stock in DMSO) (15 µM) or cycloheximide (50 mM stock in DMSO) (50 µM). Control treatments contained an equal amount of solvent. Double drug treatments were carried out with 30 or 60 min of pretreatment followed by concomitant drug treatment for the indicated times.

## GUS staining procedures and histological analysis

Plants were treated with 90% acetone on ice for 30 min, then washed twice in GUS staining buffer (50 mM sodium phosphate (pH 7), 0.1% Triton X-100, 1 mM of each $K_3Fe^{III}(CN)_6$ and $K_4Fe^{II}(CN)_6$) and stained at 37°C in darkness in GUS staining buffer containing X-Gluc in a final concentration of 0.2 mg/mL. Clearing of root tissues was done as described previously (*Malamy and Benfey, 1997*).

## Auxin transport assay

Measurement of root acropetal auxin transport was essentially done as described by (*Lewis and Muday, 2009*). In brief, 6-day-old *emo1*-RNAi seedlings were transferred to MS, 1% sucrose plates plus or minus 0.1% ethanol. Warm agar (1.25% (w/v), 50°C) was mixed with 200 nM radioactive indole-3-acetic acid (5-[3]H IAA, 20 Ci/mmol, American Radiolabeled Chemicals) and droplets (5 µL)

were prepared. Such droplets were applied exactly 1 cm above the root tip. The IAA transport was measured in vertically oriented plants (inverted plates) after 18 hr in the dark (to minimize IAA degradation) at room temperature. Therefore, the first 0.5 cm of the root underneath the site of auxin application was discarded, and the remaining root segment was used for the measurement of radioactivity.

### RNA isolation and RT-PCR

Total RNA was extracted from seedlings using the Nucleo Spin RNA II kit (Macherey-Nagel) and used as a template for cDNA synthesis with RevertAid H Minus Reverse Transcriptase (Thermo Scientific, Germany) according to the manufacturer's instructions. A complete list of oligonucleotides used for PCR and RT-PCR is provided below.

### Preparation of microsomal membranes

Microsomal membrane fractions were prepared from 7 day-old seedlings. Tissue was homogenized with 3.5 mL homogenization buffer per mg FW (330 mM sucrose, 50 mM Tris-HCl (pH 7.5), 0.5% (w/v) casein, 1.5% (w/v) PVP-40, 3 mM DTT, 5 mM EDTA and Complete protease inhibitor mixture (Roche)). The homogenate was centrifuged at 10,000 g for 20 min at 4°C. The supernatant was centrifuged at 110,000 g for 45 min at 4°C. The microsomal pellet was resuspended in 50 mM Tris-HCl (pH 7.5),10 % (v/v) glycerol, 1 mM EDTA, 1 mM DTT and Complete protease inhibitor mixture.

### Yeast two-hybrid assay, SDS-PAGE and Western blotting

For yeast two-hybrid analyses, the individual constructs were cloned into the vectors pGADT7 and pGBKT7 and co-transformed into the yeast strain PJ69-4A. The PIN2 hydrophilic loop region analyzed comprises amino acids 189–477. Activity of the *ADE2* reporter was analyzed by growth of transformed yeast on SD medium lacking adenine. SDS-PAGE, Western blotting and immunodetection followed standard procedures.

### CoIP-MS analysis

*Arabidopsis* seedlings expressing 14-3-3 epsilon-GFP (endogenous promoter) and, as control, GFP (UBQ10 promoter) were grown under continuous light in liquid medium. Three grams plant tissue were taken for the immunoprecipitation as described in (*Park et al., 2012*) with slight modifications. The lysis buffer contained 1% Triton X-100 and PhosSTOP phosphatase inhibitor cocktail (Roche). GFP-Trap Beads (50 µL) (ChromoTek) were added to each sample and the final precipitate in Laemmli buffer was analyzed by MS at the University of Tübingen Proteome Center. Following a tryptic in gel digestion, LC-MS/MS analysis was performed on a Proxeon Easy-nLC coupled to an Orbitrap Elite mass spectrometer (method: 130 min, Top15, HCD). Processing of the data was conducted using MaxQuant software (vs 1.5.2.8). The spectra were searched against an *A. thaliana* Uni-Prot database. Raw data processing was performed with 1% false discovery rate setting.

Two individual biological replicates were performed and the following candidates were omitted from the list of epsilon-GFP interaction partners: (i) proteins that interacted with GFP (control), (ii) proteins that were identified in only one of the two experiments and (iii) mitochondrial and chloroplastic proteins.

### Ratiometric bimolecular fluorescence complementation (rBiFC)

14-3-3 isoforms and the PIN2 hydrophilic loop region were amplified from existing templates and cloned, using GATEWAY technology (Life Technologies), into pBiFCt-2in1_CC allowing for (i) simultaneous cloning of two genes into the same vector backbone and (ii) ratiometric analysis due to additional expression of mRFP (*Grefen and Blatt, 2012*). Fluorescence intensity (reconstituted YFP vs. RFP) was measured after transient *Agrobacterium*-mediated transformation of *Nicotiana benthamiana* leaves as described (*Grefen and Blatt, 2012*).

### Fluorescent protein constructs and fluorescence microscopy

The genomic sequence of diverse 14-3-3 isoforms, comprising the respective promoter (intergenic region), was obtained from BAC clones either by restriction enzyme digestion and/or by PCR-based amplification and finally ligated into pGTkan (*Dettmer et al., 2006*). All PCR-amplified fragments

were controlled by sequencing. Transgenic plants were selected based on the kanamycin resistance conferred by pGTkan and homozygous lines were established.

Live-cell imaging was performed using 4- to 6-day-old seedlings and the Leica TCS SP8 confocal laser scanning microscope. FM4-64 was used at 2 μM final concentration. The excitation wavelength was 488 nm, and the emission was detected for GFP between 500 and 530 nm and for FM4-64 between 620 and 680 nm. All CLSM images in a single experiment were captured with the same settings using the Leica Confocal Software. All the experiments were repeated at least three times. Images were processed using Adobe Photoshop CS3. Quantification of the fluorescence signal was performed by Image J software. For distribution of DR5rev::GFP, signals in lateral root cap cells of gravistimulated roots (proximal to quiescent center) were assessed. The basal/lateral ratio of signal intensities of PIN-GFP proteins was calculated by determination of mean values of defined PM-areas at the basal and lateral side of root endodermis (PIN1-GFP) or cortex (PIN2-GFP) cells. For the evaluation of sensitivity to BFA, the mean fluorescence intensity of PIN2-GFP was measured in defined areas of BFA bodies and the PM. Washout efficiency was calculated on the basis of the BFA/PM intensity ratio before and after washout of BFA.

## Analysis of publicly available expression data and phylogenetic relationship of 14-3-3 family members

Multiple alignment of full-length protein sequences of published genes encoding for 14-3-3 isoforms in *A. thaliana* (Sehnke et al., 2002), *S. lycopersicum* (Xu and Shi, 2006), *O. sativa* (Yao et al., 2007), *M. truncatula*, *P. trichocarpa* and *P. patens* (Tian et al., 2015) was performed using CLC Main Workbench 7.8. A maximum likelihood phylogenetic tree was constructed with 10,000 bootstrap replicates.

The normalized microarray data provided in the 'Developmental Map' of the eFP Browser at the Bio-Analytic Resource for Plant Biology (BAR) (http://bar.utoronto.ca) (Arabidopsis eFP Browser, Medicago eFP Browser, Poplar eFP Browser, Rice eFP Browser, Physcomitrella eFP Browser) and the RNA-Seq data available at the Tomato Functional Genomics Database (http://ted.bti.cornell.edu/cgi-bin/TFGD/digital/home.cgi) (Digital expression experiment D006: Transcriptome analysis of various tissues in wild species *S. pimpinellifolium*) were used for the tissue-specific expression analysis. Absolute signal intensity values for probe set IDs corresponding to 14-3-3 genes as well as ubiquitin were selected for further analyses.

## List of primers used in this study

| | |
|---|---|
| epsilon-RNAi-F | TAT<u>GGATCCCTCGAG</u>CTATCGCTATCTTG |
| epsilon-RNAi-R | TATA<u>GTACTCCCGGGG</u>GGTGCCAAACCATTCTC |
| mu-RNAi-F | TATA<u>GGCCTGATATC</u>AAAGGAAGCAGTGAAAGG |
| mu-RNAi-R | TATA<u>TCGATGGTACC</u>ATTCACCCTCGGAAGCCG |
| omicon-RNAi-F | TATA<u>GTACTCCCGGG</u>TTTCAGATACTTGGCTGA |
| omicon-RNAi-R | TATA<u>AGCTTGAATTC</u>TGTAGACAATTCCGTGCT |
| eps_mu_miR-F | gaTTGAACTCCGCAATATAGCGGtctctcttttgtattcc |
| Ep_mu_miR-R | gaCCGCTATATTGCGGAGTTCAAtcaaagagaatcaatga |
| Ep_mu_miR*F | gaCCACTATATTGCGCAGTTCATtcacaggtcgtgatatg |
| Ep_mu_miR*R | gaATGAACTGCGCAATATAGTGGtctacatatatattcct |
| Om_miR-F | gaTCAAATGAATGTATGGCGCCGtctctcttttgtattcc |
| Om_miR-R | gaCGGCGCCATACATTCATTTGAtcaaagagaatcaatga |
| Om_miR*F | gaCGACGCCATACATACATTTGTtcacaggtcgtgatatg |
| Om_miR*R | gaACAAATGTATGTATGGCGTCGtctacatatatattcct |
| eps-RT-F | CTATCGCTATCTTGCGGA |
| eps-RT-R | GTCGACTTAGTTCTCATCTTGAGG |
| mu-RT-F | ATCTTGAAAATCAGTCATGGGTTCT |

| mu-RT-R | ATTCACCCTCGGAAGCCG |
|---|---|
| mu-g-F | GTTGTTATGTTACCATATCTA |
| mu-g-R | AGACCTGATTCGAGTGAGAAG |
| omicron-RT-F | GAACGAGAGAGCGAAGCAAGTG |
| omicron-RT-R | TGTAGACAATTCCGTGCT |
| AHA2-RT-F | ACACAAAGACGCAAACCTC |
| Rbcs-3A-R | TTTATTAACTCTTATCCATCCATTTGC |
| Actin-F | TCCAAGCTGTTCTCTCCTTG |
| Actin-R | GAGGGCTGGAACAAGACTTC |

## Acknowledgements

We are very grateful to Sarah Naumann for performing excellent experiments and analyzing data. We thank Gerd Jürgens for critical reading of the manuscript as well as for providing seeds of transgenic Arabidopsis plants, Christian Luschnig for providing PIN2 antiserum, Sandra Richter, Misoon Park and Manoj Singh for SP8 support, Andrea Bock for technical support as well as Sascha Laubinger and Chang Liu for help with the expression analysis. MS analysis was done at the Proteome Center, University of Tübingen, and we thank Irina Droste-Borel and Mirita Franz-Wachtel for their help in interpreting the data. This work was funded by the DFG.

## Additional information

### Funding

| Funder | Grant reference number | Author |
|---|---|---|
| Deutsche Forschungsgemeinschaft | OE 205/1-1 | Claudia Oecking |
| Deutsche Forschungsgemeinschaft | OE 205/2-1 | Claudia Oecking |

The funders had no role in study design, data collection and interpretation, or the decision to submit the work for publication.

### Author contributions

JK, Data curation, Formal analysis, Validation, Investigation; NJ, Supervision, Investigation, Visualization, Writing—review and editing; KW, CM, CT, Data curation, Formal analysis, Validation, Investigation, Visualization; AK, Investigation; CO, Conceptualization, Formal analysis, Supervision, Funding acquisition, Writing—original draft, Project administration, Writing—review and editing

### Author ORCIDs

Claudia Oecking, http://orcid.org/0000-0003-0635-6457

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
