## [Decision Letter]

[Editors’ note: this article was originally rejected after discussions between the reviewers, but the authors were invited to resubmit after an appeal against the decision.]

Thank you for submitting your work entitled "Arabidopsis 14-3-3 epsilon members contribute to polarity of PIN auxin carrier and auxin transport-related development" for consideration by *eLife*. Your article has been favorably evaluated by a Senior Editor and two reviewers, one of whom is a member of our Board of Reviewing Editors. The reviewers have opted to remain anonymous.

Our decision has been reached after consultation between the reviewers. Based on these discussions and the individual reviews below, we regret to inform you that your work will not be considered further for publication in *eLife*.

The authors sought to establish roles for the epsilon group of 14-3-3 proteins. The work is nicely carried out and the cell biology experiments are convincing. However, in the absence of any plausible target(s) and/or the protein kinase(s)/phosphatase(s) involved it is difficult to rule out indirect effects.

Reviewer #1:

14-3-3 proteins interact with phosphorylated versions of multiple client proteins in order to affect the activity state of the clients, or to change the subcellular location of the clients. There are two type of 14-3-3 proteins, epsilon and non-epsilon. Non-epsilons are thought to have specific roles (sometimes specific to organisms?), and the epsilons, being basal, are thought to do something fundamental, but it is unclear what, since there is gene redundancy and multiple knockouts have not been revealing. The authors used inducible RNAi to dissect the roles of three epsilon type 14-3-3 members in Arabidopsis seedlings (the other 2 members are not expressed there). They found some seedling phenotypes that pointed to auxin problems, and they show that an obvious guess (H^+^ ATPase, a well-known interactor of 14-3-3s) does not explain the phenotypes. Another possibility was PINs (auxin transporters), but they didn't find evidence for direct 14-3-3 interactions with PINs – however, they did show that PINs are mis-localized in 14-3-3 RNAi lines, and they showed that 14-3-3 RNAi lines have problems with membrane trafficking, by performing the expected (routine) experiments with endosome markers and pharmacological inhibitors. I have to say that the cover letter implied a more exciting conclusion than what I ended up with, after reading the paper, and even their cover letter ending is a bit anti-climactic, saying that these epsilons "indeed fulfill basal cellular functions". So, my feelings are mixed, I think the experiments are sound but I am not sure how exciting they are. Maybe I would be more excited if I got a better sense of the evolutionary story about 14-3-3s across all plants; there is a lot of expression data for many plant genomes – how widespread are these Arabidopsis-centric findings likely to be?

Reviewer #2:

In this work the expression in Arabidopsis of individual isoforms within the epsilon group of 14-3-3 proteins have been modified and interesting phenotypes are described that relate to changed growth, auxin responses, subcellular polarity of PIN proteins and intracellular membrane trafficking. The work is carried out carefully and is of high quality. Unfortunately, the molecular mechanism underlying the described phenotypes remains unclear. It remains therefore uncertain whether the effects are direct or indirect, or via one or multiple 14-3-3 protein targets. Some possible targets are discussed but none have been identified.

It is a long-standing discussion in the field whether isoform diversity explains functional differences in the 14-3-3 protein family. The number of targets of 14-3-3 proteins is in the hundreds whereas the number of 14-3-3 protein isoforms is much less, raising doubts about the specificity of 14-3-3 proteins, which is pronounced by the fact that 14-3-3 proteins work across kingdoms (fx brain 14-3-3 works well on plant protein targets not found in animals). This would suggest that it is not the 14-3-3 proteins themselves that deliver specificity, but rather it is specific protein kinases and phosphatases that dictate whether the 14-3-3 proteins bind or not to a specific target. One reasonable model is that they provide a means for individual cell types to control the expression of 14-3-3 proteins during development and depending on its specific needs. The manuscript would gain from referring the results to this discussion.

I find no evidence presented for the functional model (i.e. specific isoforms have specific functions) although it seems to be taken for granted that epsilon members have such specific functions. Rather, in the present work, it is clearly shown that the tissue-specific expression is unique for each individual epsilon member. Is this in accordance with the genetic model for isoform diversification (i.e. the purpose of gene duplications is a means to allow for diversification in gene promoters rather than in the protein sequences)?

[Editors’ note: what now follows is the decision letter after the authors submitted for further consideration.]

Thank you for resubmitting your work entitled "Arabidopsis 14-3-3 epsilon members contribute to polarity of PIN auxin carrier and auxin transport-related development" for further consideration at *eLife*. Your revised article has been favorably evaluated by Ian Baldwin (Senior Editor), a Reviewing Editor, and one reviewer.

The manuscript has been improved but there are some remaining issues that need to be addressed before acceptance, as outlined below:

You did not adequately address the query: "there is a lot of expression data for many plant genomes – how widespread are these Arabidopsis-centric findings likely to be?". All that was added to the revised manuscript is a sentence stating that such genes are found in all plants, which does not address expression patterns. We are not expecting you to study other plant species experimentally, but a meta-analysis of other published or publicly accessible RNA-seq or microarray data is certainly feasible, as is a phylogenetic analysis of this protein family. This might make the experimental findings in Arabidopsis relevant to a broader range of scientists.

Regarding the Co-IP and Mass Spectrophotometry experiment – Table 1 is lacking in detail, i.e. statistical analysis should be provided, as well as more details about how many peptides for a given protein were detected, criteria for inclusion, etc. To our knowledge, it is not typical to include proteins found in only one of two Co-IP experiments, and restricting the list to only membrane trafficking proteins is also unexpected and biased (see reviewer comments).

Reviewer #2:

In the revised version of this manuscript, potential targets of 14-3-3 epsilon-GFP have been identified by co-immunoprecipitation experiments coupled with mass spectrometry-based protein identification. Multiple potential targets were identified including a large number of proteins involved in membrane trafficking. A direct phospho-dependent interaction between 14-3-3 epsilon proteins and these proteins provides a possible explanation for the trafficking defects observed in 14-3-3 epsilon mutants.

The list of 14-3-3 protein interactors (Table 1) has been restricted to proteins with a role in membrane trafficking. This is unfortunate, as this selection might be biased. Did the experiment account for membrane proteins? In particular, the reader would like to know whether the plasma membrane H^+^-ATPase and/or PIN proteins were co-immunoprecipitated with 14-3-3 epsilon protein or not.

---

## [Author Response]

[Editors’ note: the author responses to the first round of peer review follow.]

The authors sought to establish roles for the epsilon group of 14-3-3 proteins. The work is nicely carried out and the cell biology experiments are convincing. However, in the absence of any plausible target(s) and/or the protein kinase(s)/phosphatase(s) involved it is difficult to rule out indirect effects.

Reviewer #1: […] I have to say that the cover letter implied a more exciting conclusion than what I ended up with, after reading the paper, and even their cover letter ending is a bit anti-climactic, saying that these epsilons "indeed fulfill basal cellular functions". So, my feelings are mixed, I think the experiments are sound but I am not sure how exciting they are.

The cover letter started with the statement that “we addressed a major, unresolved problem in biology, namely the physiological role of functionally redundant members of the 14-3-3 protein family in higher organisms”. This is exactly what we did by generating Arabidopsis lines allowing for conditional genetic inactivation of three 14-3-3 isoforms belonging to the ancestral epsilon group which is proposed to possess *basic*, fundamental eukaryotic functions. According to the reviewer, the cover letter ended *‘anti-climactic’* by stating that the epsilon members indeed fulfill *basal* cellular functions. Based on our studies, we unambiguously could link 14-3-3 epsilon activities to defined post-Golgi trafficking processes regulating – amongst others – polar transport of the phytohormone auxin. Beyond any doubt, this is a process of fundamental importance.

Maybe I would be more excited if I got a better sense of the evolutionary story about 14-3-3s across all plants; there is a lot of expression data for many plant genomes – how widespread are these Arabidopsis-centric findings likely to be?

14-3-3 isoforms belonging to the epsilon group are present in all plant genomes sequenced so far. Such a statement is now integrated into the revised manuscript (Introduction, first paragraph).

Reviewer #2:

In this work the expression in Arabidopsis of individual isoforms within the epsilon group of 14-3-3 proteins have been modified and interesting phenotypes are described that relate to changed growth, auxin responses, subcellular polarity of PIN proteins and intracellular membrane trafficking. The work is carried out carefully and is of high quality. Unfortunately, the molecular mechanism underlying the described phenotypes remains unclear. It remains therefore uncertain whether the effects are direct or indirect, or via one or multiple 14-3-3 protein targets. Some possible targets are discussed but none have been identified.

We now integrated biochemical data based on immunoprecipitation of interactors of 14-3-3 epsilon-GFP (GFP served as negative control) as identified by MS-based protein identification. Several proteins known to be involved in cellular trafficking processes co-precipitated with the 14-3-3 epsilon isoform, among those key players, such as small GTPases and their regulators (Table 1, text: subsection “14-3-3 epsilon group members may directly regulate trafficking processes”). Beside the fact that epsilon group members localize to post Golgi compartments, this provides additional evidence that 14-3-3 proteins regulate vesicle trafficking directly. A more detailed functional analysis of these interactors (most of which are still uncharacterized) will, however, be subject to future, substantial experimentation.

[Editors’ note: the author responses to the re-review follow.]

The manuscript has been improved but there are some remaining issues that need to be addressed before acceptance, as outlined below:

You did not adequately address the query: "there is a lot of expression data for many plant genomes – how widespread are these Arabidopsis-centric findings likely to be?". All that was added to the revised manuscript is a sentence stating that such genes are found in all plants, which does not address expression patterns. We are not expecting you to study other plant species experimentally, but a meta-analysis of other published or publicly accessible RNA-seq or microarray data is certainly feasible, as is a phylogenetic analysis of this protein family. This might make the experimental findings in Arabidopsis relevant to a broader range of scientists.

We have now performed a phylogenetic analysis of 14-3-3 family members from six plant species (*Arabidopsis thaliana, Solanum lycopersicum, Medicago truncatula, Populus trichocarpa, Oryza sativa* and *Physcomitrella patens*) (Figure 1—figure supplement 1). Furthermore, we have analyzed tissue-specific expression patterns of 14-3-3 encoding genes from these plant species based on publicly available microarray or RNA-seq data ([Supplementary-material SD1-data]). The ‘Materials and methods’ section has been extended accordingly (subsection “Analysis of publicly available expression data and phylogenetic relationship of 14-3-3 family members”). The corresponding paragraph within the ‘Introduction’ reads now as follows:

“Members of both groups are present in all plant genomes sequenced so far. The phylogenetic relationship of 14-3-3 family members from six plant species (*A. thaliana, Solanum lycopersicum, Medicago truncatula, Populus trichocarpa, Oryza sativa, Physcomitrella patens*) and their expression patterns (based on publicly accessible RNA-seq or microarray data) are depicted in Figure 1—figure supplement 1 and [Supplementary-material SD1-data], respectively. Except for the moss, *P. patens*, the total transcript level of all non-epsilon members in a given plant species is generally higher than that of the epsilon group isoforms. However, whether this relates to lower amounts of 14-3-3 epsilon proteins in individual cells or tissues remains unclear.”

Regarding the Co-IP and Mass Spectrophotometry experiment – Table 1 is lacking in detail, i.e. statistical analysis should be provided, as well as more details about how many peptides for a given protein were detected, criteria for inclusion, etc. To our knowledge, it is not typical to include proteins found in only one of two Co-IP experiments, and restricting the list to only membrane trafficking proteins is also unexpected and biased (see reviewer comments).

We now present the complete list of 14-3-3 epsilon interactors including the number of identified peptides, sequence coverage as well as intensities as source data file ([Supplementary-material SD2-data]). Both the complete list of interactors and the list of membrane trafficking proteins given in Table 1 have been restricted to proteins identified in two out of two CoIP-MS experiments. We furthermore expanded the ‘Materials and methods’ section for more details and statistical analysis (subsection “CoIP-MS Analysis”).